# Neuroinflammation in Friedreich’s Ataxia

**DOI:** 10.3390/ijms23116297

**Published:** 2022-06-04

**Authors:** Savina Apolloni, Martina Milani, Nadia D’Ambrosi

**Affiliations:** Department of Biology, Tor Vergata University of Rome, 00133 Roma, Italy; martina.milani@uniroma2.it

**Keywords:** frataxin, microglia, astrocytes, neurons, iron

## Abstract

Friedreich’s ataxia (FRDA) is a rare genetic disorder caused by mutations in the gene frataxin, encoding for a mitochondrial protein involved in iron handling and in the biogenesis of iron−sulphur clusters, and leading to progressive nervous system damage. Although the overt manifestations of FRDA in the nervous system are mainly observed in the neurons, alterations in non-neuronal cells may also contribute to the pathogenesis of the disease, as recently suggested for other neurodegenerative disorders. In FRDA, the involvement of glial cells can be ascribed to direct effects caused by frataxin loss, eliciting different aberrant mechanisms. Iron accumulation, mitochondria dysfunction, and reactive species overproduction, mechanisms identified as etiopathogenic in neurons in FRDA, can similarly affect glial cells, leading them to assume phenotypes that can concur to and exacerbate neuron loss. Recent findings obtained in FRDA patients and cellular and animal models of the disease have suggested that neuroinflammation can accompany and contribute to the neuropathology. In this review article, we discuss evidence about the involvement of neuroinflammatory-related mechanisms in models of FRDA and provide clues for the modulation of glial-related mechanisms as a possible strategy to improve disease features.

## 1. Introduction

Friedreich’s ataxia (FRDA) is a recessive genetic disorder involving mainly the nervous system, caused by mutations in the gene frataxin (*FXN*), which is located in the centromeric region of chromosome 9q. When the most common mutation, which consists of the expansion of GAA repeats in the first *FXN* intron, is present in homozygosis, it leads to the decreased transcription of *FXN*, with pathologic loss of the protein function. The severity of the disease depends on the number of repeats, and it can be associated with progressive ataxia, weakness, and sensory deficits, with symptom onset usually occurring in childhood or adolescence. Patients often die of cardiomyopathy. The disease has an incidence of one case in 50,000 individuals in the Caucasian population, and represents the most common of the hereditary ataxias [1].

*FXN* encodes for a ubiquitous mitochondrial protein involved in iron homeostasis and metabolism, and in the biogenesis of iron−sulphur (Fe-S) clusters, thereby affecting electron transport chain activities and several metabolic, DNA repair, and ribosome biogenesis enzymes [2]. Consequently, in FRDA, the Fe−S cluster formation is decreased [3,4], and iron levels regulatory systems (e.g., iron regulatory proteins, transferrin receptor, and ferritin) are dysregulated. FXN downregulation results in mitochondrial dysfunction, decreased ATP production, oxidative stress, free radical accumulation, and consequent cell death. Indeed, upon iron homeostasis loss, free iron can increase and interact with oxygen molecules to form reactive oxygen species (ROS) through the Fenton and Haber−Weiss reactions [5,6]. Principal damage occurs in these tissues where FXN levels are high, such as the heart, liver, skeletal muscles, and spinal cord [7]. Even though FXN deficiency is ubiquitous, FRDA shows specific neurological deficits involving different subsets of neurons. The most affected areas of the nervous system are the dorsal root ganglia, corticospinal tract, ventral/dorsal spinocerebellar tracts, and the cerebellar dentate nucleus [8].

Moreover, although the pathological manifestations of FRDA are primarily observed in the neurons of both the central and peripheral nervous system, it is emerging that the surrounding non-neuronal cells may also contribute to the pathogenesis of the disease, as demonstrated for other neurodegenerative disorders [9]. The involvement of non-neuronal cells can be attributed to the “dying back” mechanism, described as progressive axon degeneration and secondary gliotic events in the spinal cord [7]. In addition, non-neuronal cells may participate in neuron death as one of the direct results of FXN loss of function and incapability to handle and buffer iron, contributing to further iron accumulation in the nervous system. [10,11]. Indeed, in physiological conditions, the glia can largely take up excessive iron from the environment in the form of the non-transferrin-bound iron (NTBI)-mediated transport pathway [12,13], a process that could be affected in FRDA [14]. Finally, because of iron dysmetabolism and electron transporters impairment, glial cells may overproduce the extracellular and intracellular reactive species responsible for their pro-inflammatory activation, damaging the neurons [15,16]. Therefore, the study of the effects of FXN deficiency in glial cells in FRDA deserves attention because of its potential implications in disease treatments.

This review will present the contribution of neuroinflammation to the FRDA pathology and will discuss how modulating this mechanism could be a valuable means to modify pathological phenotypes.

## 2. Neuroinflammation and Iron Dysmetabolism

In recent years, neuroinflammation in neurodegenerative disorders has been assumed to be of great importance, and iron accumulation has been identified as a significant driver of cell reactivity in the brain [17]. Nervous system-resident glia (mainly microglia and astrocytes), endothelial cells, and infiltrating immune cells produce cytokines, chemokines, ROS, and little messengers in response to an alteration to the nervous tissue homeostasis. While this response is necessary to protect the nervous system from pathogens and damaging events, it substantially contributes to diseases when it becomes a chronic process that is incapable of auto-extinguishing [18].

Neuroinflammation often precedes and worsens overt neuron death. In different neurodegenerative conditions with a robust neuroinflammatory component, contrasting neuroinflammation has been proven to be one of the most efficient ways to slow down disease progression [9]. The microglia are considered to be the immune cells of the central nervous system (CNS), constantly surveilling their surroundings and reacting to any perturbation in their environment. Microglial activation is often described as classical M1 or as alternative M2, similar to the nomenclature used for macrophages, but these terms oversimplify the complex process of microglial activity, which is quite variable and context-dependent [19]. Microglia are the primary producers of ROS in the CNS, mainly via the enzyme NADPH oxidase 2 (NOX2). Indeed, oxidative stress and reactivity in these cells are indissoluble events that can progressively lead to a harmful cytotoxic phenotype [20,21]. In fact, besides being directly toxic to neurons, as cell-to-cell messengers, ROS can cause a noxious imbalance in the crosstalk between neurons and the surrounding cells [22]. During adaptive responses by immune cells, the release of reactive species is a controlled physiological process necessary to spread an efficient response and clear the source of damage. However, if the equilibrium between the formation of reactive species and the endogenous antioxidant defense is altered, as it occurs in FRDA [23], ROS release by glial cells can cause further oxidative stress and foster damage. In addition to ROS imbalance, recent data demonstrate that iron overload in the microglia can promote their polarization towards the pro-inflammatory phenotype, leading to the release of stress-associated cytokines that further increase inflammation, resulting in a loop process [24]. Furthermore, microglial iron transport pathways are differentially active in response to pro- and anti-inflammatory stimuli [25]. Therefore, iron dysregulation in the microglia is tightly linked to oxidative stress and neuroinflammation, and compounds promoting iron sequestration can suppress microglia activation in acute and chronic nervous system injuries [26,27]. However, besides their cytotoxic role, it is emerging that microglia activity is essential for remyelination, proper synaptic pruning, and overall correct circuitry functioning [28,29]. Iron homeostasis is one essential condition for these processes and its alteration could impair these necessary physiological functions.

Astrocytes constitute the scaffold of the entire CNS, and their processes participate in the neurovascular unit of the blood−brain barrier (BBB). Their functions range from regulating cerebral blood flow, to maintaining water, small molecules, and neurotransmitter homeostasis, forming synapses, and supporting neuron metabolism. Like microglia, astrocytes respond to insults through reactive astrogliosis that, if sustained, can result in the formation of a glial scar as part of the neuroinflammatory process [30]. Because of their proximity to the blood vessels, astrocytes are crucial for iron transport into the brain. Indeed, astrocytes are the brain cells that display the highest expression of hepcidin, the master regulator of iron homeostasis. It has been proposed that hepcidin secreted by astrocytes can act on ferroportin (FPN1) and regulate iron intake at the BBB, acting on the brain microvascular endothelial cells [12]. In a neurodegenerative context characterized by brain iron accumulation and neuroinflammation, such as for Alzheimer’s disease (AD), hepcidin derived from astrocytes can ameliorate AD symptoms of APP/PS1 transgenic mice, attenuating iron deposition, with consequent alleviated oxidative stress and neuronal cell death [31,32]. Therefore, these results suggest that astrocytes may participate in iron accumulation and aberrant redistribution in other disorders, including FRDA.

Therefore, iron accumulation, mitochondria dysfunction, and ROS production, which are mechanisms identified as etiopathogenic in neurons in FRDA [33], can similarly affect glial cells [8], leading them to assume phenotypes that can concur to and exacerbate neuron loss. Evidence of the involvement of neuroinflammatory mechanisms in FRDA has been detected in patients and disease models, and is summarized in Table 1.

## 3. Neuroinflammation in FRDA

Specific pathways related to neuroinflammation are altered in the microglia, astrocytes, and myelinating glial cells in FRDA, as shown in Figure 1 and reported in detail in the following paragraphs.

### 3.1. Microglia in FRDA

The first study describing glial activation in the brain of FRDA patients was reported by Koeppen et al., who observed a loss of juxtaneuronal ferritin-containing oligodendroglia and a prominent ferritin immunoreactivity in microglia and astrocytes in the dentate nucleus of patients with FRDA. These features were accompanied by neuronal atrophy and a so-called grumose degeneration, an unusual proliferation of synaptic terminals in the dentate nucleus. This degeneration indicated mitochondrial iron dysmetabolism in the terminals of corticonuclear fibers [35]. In addition, in the atrophic dentate nucleus grey matter of FRDA patients, positive ferritin microglia appeared abundant and hypertrophic, indicating an overall active state. Therefore, grumose degeneration in FRDA seemed to stimulate a prominent microglial response involving the antioxidant enzyme SOD1. This observation suggests that the accumulation of iron in FRDA generated a defensive reaction in glial cells finalized to protect neurons from oxidation [38]. Accordingly, the microglial morphology in an FRDA case showed cells with both enlarged perikarya and thicker processes [36]. The augmented microgliosis in FRDA brains could reflect the increased necessity to remove toxins, debris, and dead ferroptotic neurons, releasing iron in the interstitium through these phagocyting cells [49]. A confirmation of overall reactive gliosis in patients came from the brain positron emission tomography analysis, using the radioligand [18 F]-FEMPA to translocator protein (TSPO), a marker of microglia and astrocyte activation and proliferation. Using this technique, it was revealed that there was increased glial activation in the brain regions implicated in FRDA neuropathology, i.e., dentate nuclei, brainstem, superior cerebellar peduncles, and cerebellar cortex in individuals with FRDA, compared with the control subjects. The augmented binding of [18 F]-FEMPA was correlated with earlier disease onset, shorter disease duration, and an increase in plasma inflammatory cytokines, among which was interleukin-6 (IL-6) in patients with FRDA, indicating that chronic neuroinflammation could be a critical pathogenic mechanism in the disease [34].

The role of microglia activation in FRDA has been described in the KIKO mouse model of the disease, where the intracerebroventricular injection of the inflammatory stimulus lipopolysaccharide (LPS) induced a greater microglial activation compared with the healthy mice. The swollen cell bodies had shortened processes, suggested that the microglia in FRDA mice are in a more activated state than the microglia in wild-type mice. Furthermore, the authors observed an increase in oxidative damage and the DNA repair proteins MUTYH and PARP-1 in the cerebellar microglia of FRDA mice. These aberrant features were attenuated through the administration of PJ34, a PARP-1 inhibitor, suggesting that microglial PARP-1 could be an important therapeutic target in FRDA. The involvement of DNA damage in the activation of FXN-deficient microglia was further confirmed in experiments in vitro on microglial cell lines, where the knockdown of FXN increased DNA damage and the expression of the DNA repair genes MUTYH and PARP-1 [40].

In other mouse model of FRDA, YG8R, the transplantation of wild-type mouse hematopoietic stem and progenitor cells (HSPCs), resulted in the amelioration of muscle weakness and locomotor deficits [50]. In the histological analysis, sensory neurons in the dorsal root ganglia (DRGs) and mitochondria in the brain, skeletal muscle, and heart appeared intact. The authors demonstrated that, in FRDA mice, transplanted HSPCs were engrafted within the brain and spinal cord as differentiated microglia, and within the DRGs, peripheral nerves, skeletal muscle, and heart as differentiated macrophages. Hence, the robust neurological phenotypic rescue observed in HSPC-transplanted YG8R mice could be partly due to replacing the FXN-deficient microglial cells with wild-type microglia. Mechanistically, microglia transferred wild-type FXN and cyclooxygenase (COX) 8 mitochondrial proteins to neurons in vivo, suggesting the existence of a novel mechanism to be investigated in FRDA [50]. Sustaining the hypothesis that FTX deficiency leads to an increase in neuroinflammation and to the production of ROS, in the cerebella of both KIKO and YG8R mice, there was an upregulation in inducible COX2 expression and activity compared with the controls, accompanied by an increase in the transcription factors activator protein 1 (AP1) and cAMP response element-binding protein (CREB), known to drive COX2 expression. In addition, the authors showed that FXN deficiency increased the reactivity of the microglia in the cerebellum of YG8R mice after LPS treatment, further indicating an increased susceptibility to inflammation compared with the healthy mice [41].

### 3.2. Astrocytes in FRDA

Astrocytes play crucial roles in the pathogenesis of several forms of ataxias, where they contribute to disease progression in a phase-specific manner and represent a new target for therapeutic approaches [51]. In cerebellar tissues of FRDA patients, marked astrogliosis of the dentate nucleus is evident, as demonstrated by ferritin positive astrocytes detected near the vessel walls [35]. Moreover, the autopsy specimens of FRDA patients showed the intrusion of CNS-derived astroglia into the dorsal roots [37]. Consistently, the levels of plasma glial fibrillary acidic protein (GFAP) are significantly higher in FRDA patients, potentially reflecting glial activation [39].

Loss of FXN is detrimental not only to neurons, but also to the normal function of astroglia; cerebellar astrocytes may contribute to FRDA clinical symptoms, showing specific vulnerability to FXN deficiency [8]. In human astrocytes in vitro, the knockdown of FXN demonstrated detrimental effects to the integrity of the mitochondria, which appeared severely swollen and punctate. Accordingly, mitochondrial superoxide formation, apoptosis-related proteins p53 and p21, and activated caspase-3 were all increased. Moreover, astrocytes lacking FXN displayed abnormal secretion of several molecules, mainly associated with cell growth, immunity, and inflammation, such as IL-6 and macrophage inflammatory protein-1 alpha (MIP-1α). Remarkably, FXN-depleted astrocytes had detrimental effects on neuron development by inducing a delay in the maturation of mouse neurons and decreased neurite length and cell branching. The reduction in these features was associated with enhanced cell death, highlighting that FXN silencing in astrocytes alters their capacity to support the development of neurons. Finally, the study confirmed that the altered mitochondrial iron homeostasis in astrocytes caused by FXN deficiency leads to an increased mitochondrial iron content that favors oxidative stress and superoxide production, contributing to the non-cell-autonomous pathological process in FRDA [52].

In accordance, the astrocytes differentiated from neural stem cells obtained from the YG8R model exhibited detrimental signs, such as the reduced activity of the Fe-S containing enzyme aconitase [45]. Alteration in bioenergetic parameters is a common pathological feature of the neurodegenerative diseases leading to neuronal dysfunction, and dysfunctional aconitase, among the other bioenergetic parameters, is a crucial factor that could promote neurodegeneration [53]. In addition, FXN-deficient astrocytes showed a reduced expression of the antioxidant enzymes SOD2 and Gpx1, resulting in increased sensitivity to oxidative stress, together with a significant reduction in the expression of several DNA mismatch repair enzymes compared with the control cells [45]. Finally, FXN knockdown increased the production of ROS in the primary mouse astrocytes [44].

Sustaining a non-cell-autonomous toxic effect of FXN in vivo, the ablation of FXN in astrocytes during development in FGKO mice (where FXN is ablated in a time-dependent manner) caused severe ataxia and early death, inducing growth and survival impairments. In contrast, the mice in which FXN was knocked out in astrocytes later in life did not give rise to apparent neurological phenotypes, indicating that developing cerebellar astrocytes are more vulnerable to the lack of FXN, and suggesting a role of astrocytes in the progression of the disease [43]. Extensive neuroinflammation has been observed in YG8R mice, where FXN loss induced increased satellite cell proliferation, extensive astrocytosis, and an influx of inflammatory OX42 (CD11b/c)-positive cells in both the spinal cord and cerebellar dentate nucleus [42]. Consistently, in the KIKO mouse model, a substantial increase in astrocytosis was detected following LPS injection in the cerebellum of FRDA compared with non-transgenic mice, suggesting an increased vulnerability to inflammation, as observed for the microglia [40].

Targeting astrocytes in models of the disease represents a promising strategy. Indeed, in FGKO and YG8R mice, treatment with insulin-like growth factor I (IGF-I), previously shown to normalize FXN levels in FXN-deficient neurons and astrocyte cultures through the Akt/mTOR signaling pathway [44], proved beneficial effects for rescuing astrocyte-associated cerebellar defects and atrophy, together with the improvement of motor performances and increment in survival [43].

In addition, in YG8R mice, treatment with granulocyte-colony stimulating factor (G-CSF) and stem cell factor (SCF) markedly reduced the extent of astrocytosis and inflammatory cell infiltration within the dorsal columns, spinocerebellar, and corticospinal tracts. The results indicate that attenuating neuroinflammation slows down the progression of the disease. The neuroprotective action of this combined treatment was indeed exerted at a clinical level, resulting in a significant improvement in motor coordination and locomotor activity, even after the onset of neurological symptoms [42].

It has recently been demonstrated that targeting Sonic Hedgehog (SHH) with the Smoothened antagonist (SAG) rescued mitochondrial dysfunction and reverted the neurotoxicity induced by the lack of FXN in human astrocytes in vitro, showing the potential of pharmacologically targeting astrocytes cells to attenuate neurodegeneration in FRDA [46].

In Drosophila melanogaster, FXN knockdown in the glia affects fly locomotion, increases brain vacuolization due to cellular degeneration, and induces defects in lipid metabolism and oxidative stress [47], suggesting a role for these cells in the pathology. The expression of Glaz, one of the Drosophila homologs of apolipoprotein D (ApoD), in the glia of FXN-deficient flies, was sufficient to increase the lifespan and improve locomotor activity, likely because of its modulation of lipid composition and oxidation [48]. Interestingly, a genetic screen in the same model identified Drosophila mitofusin (Marf), a gene involved in mitochondrial fusion and degradation, as lying at the interface between the mitochondria and endoplasmic reticulum, a critical mediator of the pathology in the glia. Marf downregulation fully rescued some of the essential phenotypes induced by FXN silencing in the glia, such as locomotor dysfunction, brain degeneration, and lipid dyshomeostasis in the brain [54].

Overall, these results demonstrate that astrocyte activation could exacerbate or even cause neuronal dysfunctions, triggering a further amplification of astrogliosis in a detrimental vicious circle.

### 3.3. Myelinating Glial Cells in FRDA

Finally, myelinating glial cells are also implicated in FRDA, with oligodendroglia and Schwann cells being highly susceptible to FXN deficiency. In the dentate nucleus of FRDA patients, ferritin is expressed mainly in the oligodendrocytes, while, as the disease progresses and neurons begin to undergo atrophy, these cells disappear, being replaced by positive ferritin microglia [35]. As demonstrated in vitro, a significant decrease in the proliferation of both oligodendrocytes and Schwann cells occurred after FXN knockdown through the activation of the inflammatory pathways. Indeed, in FXN-deficient Schwann cells, the microarrays analysis showed a decrease in antioxidant genes and a substantial increase in inflammatory genes, such as IL-1β, IL-1α, IL-6, NFκB1, and Tumor Necrosis factor, confirmed at both the mRNA and protein levels, suggesting that the inflammatory cytokines produced by these cells may contribute to DRG neuron loss [11].

Altogether, this evidence suggests that the expression of mutant FXN in glial cells may act as a trigger, responsible for their reprogramming and functional impairment, contributing to the degeneration of nearby neurons.

## 4. Conclusions

FRDA is a chronic, progressive, and frequently life-threatening neuromuscular disease. Like all rare diseases, it has a high impact on the quality of life of patients and their families, accompanied by the lack of public awareness.

Unfortunately, ataxia has no cure, and the current therapies are aimed at motor re-education or muscular reinforcement. Several potential treatments have been subjected to clinical trials or are being developed for human studies, and include strategies that increase FXN levels, protein and gene replacement therapies, antioxidants, iron chelators, and modulators of inflammation [55]. The absence of an effective treatment for FRDA is mainly due to underestimating some processes influencing denervation-induced muscle atrophy.

Among these mechanisms, the contribution of neuroinflammation to the pathology has been largely neglected, possibly missing out on relevant diagnostic, prognostic, and therapeutic targets of the disease. Recent findings suggest that the inflammatory response in the cerebellum and spinal cord may be a critical mechanism in the pathogenesis of FRDA, and that modulating neuroinflammation could be a possible strategy for the control of the disease.

The studies discussed in this review support a strong involvement of the neuroinflammatory mechanisms in FRDA, which has both pathomechanistic and therapeutic implications (Table 1). Studies of people with FRDA and animal and cell models have provided much insight into the pathogenesis of this disorder, but it is still to be defined whether neuroinflammation is a cause or consequence of disease onset. Glial activation indeed reflects both a response to neuronal loss and the direct result of FXN knockdown that has been shown to cause neuroinflammation, producing a cytotoxic environment [8]. The role of FXN in influencing glia activity and damaging neurons indicates a non-cell-autonomous mechanism in FRDA.

In conclusion, we believe that investigating the role of neuroinflammation is fundamental for the comprehension and manipulation of the progression of FRDA. Glia targeting could play a valuable role in ameliorating neuronal circuits in FRDA-affected CNS regions, consistently with other neurodegenerative conditions, where the modulation of inflammation represents one of the most promising therapeutic strategies.

## Figures and Tables

**Figure 1 ijms-23-06297-f001:**
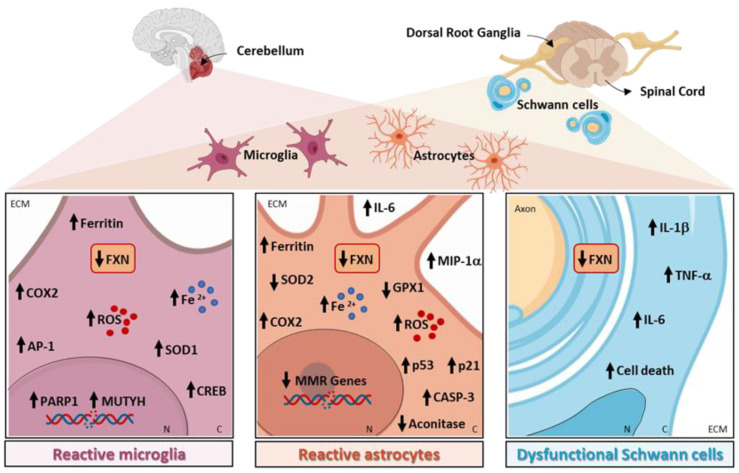
Neuroinflammation-related pathways altered in Friedreich’s ataxia (FRDA). Cerebellum, spinal cord, and dorsal root ganglia are three of the principal nervous system organs involved in the pathogenesis of FRDA. The genetically decreased expression of frataxin (FXN) leads to the disturbance of the metabolism of iron with the consequent iron increase in reactive microglia and astrocytes, together with mitochondria dysfunctions. FRDA microglia show an increase in oxidative damage, and the DNA repair proteins MUTYH and PARP-1, reactive oxygen species (ROS), activator protein 1 (AP1), and cAMP response element-binding protein (CREB), known to drive cyclooxygenase 2 (COX2) expression. In FRDA astrocytes, the depletion of FXN leads to an increase in ROS, COX2, MIP-1α, IL-6, p53, p21, and activated caspase-3 (CASP3), and to a decrease in mitochondrial aconitase, Pgc-1α, Sod2, and glutathione peroxidase 1 (GPX1) with a significant reduction in the expression of several MMR genes. Finally, FXN deficiency causes an increase in IL-6, IL-1b, and TNF-a in dysfunctional Schwann cells. N, nucleus; C, cytosol; ECM, extracellular matrix.

**Table 1 ijms-23-06297-t001:** Evidence for the involvement of neuroinflammation in Friedreich’s ataxia.

FRDA Model	Neuroinflammatory Features	References
**Patients**	Increased glial activation in cerebellum and brainstem	[34]
	Increased ferritin signals in cerebellar microglia and astrocytes	[35]
	Microglia with enlarged perikarya and thicker processes	[36]
	Astroglia intrusion into dorsal roots	[37]
	Hypertrophic cerebellar microglia positive for SOD1 enzyme	[38]
	Increased GFAP plasma levels	[39]
**KIKO mice**	Increased cerebellar microgliosis and astrocytosis after LPS stimulation; increased oxidative damage and DNA repair proteins	[40]
	Increased cerebellar COX2	[41]
**YG8R mice**	Increased cerebellar microglial activation after LPS; increased COX2	[41]
	Increased satellite cell proliferation, astrocytosis and influx of OX42 positive cells in the spinal cord and cerebellum	[42]
**FGKO mice**	Severe ataxia after frataxin deletion in astrocytes during development	[43]
**Microglial cell lines**	Increased DNA damage after frataxin knockdown	[40]
**Mouse primary astrocytes**	Increased ROS production after frataxin knockdown	[44]
**Human astrocytes** **in vitro**	Impaired mitochondrial activity and superoxide formation; increased release of inflammatory molecules and toxicity for neurons after frataxin knockdown	[45,46]
**iPSC-derived YG8R astrocytes**	Reduced aconitase and DNA repair enzymes; increased sensitivity to oxidative stress	[45]
**Schwann cells** **in vitro**	Decreased proliferation and increased inflammatory genes after frataxin knockdown	[11]
* **Drosophila** * * **melanogaster** *	Locomotor dysfunction, brain degeneration and lipid metabolism defects after frataxin knockdown in glia	[47,48]

## Data Availability

Not applicable.

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
