# Peer review of "Neuroinflammation in Friedreich’s Ataxia"

_ijms, 2022, doi:10.3390/ijms23116297_

Round 1

Reviewer 1 Report

1.  The manuscript must be reviewed for English language grammar and style, as some errors were present, and some parts are difficult to read/understand.

2.    Please define acronyms at their first use (e.g. CNS)

3.    Gene/protein nomenclature should be used following standard rules. i.e. FXN gene, FXN protein.

4.    Please use acronyms homogeneously in the manuscript, i.e. line 47, frataxin level, and line 48, FXN deficiency.

5.    Some conclusions derived from the reviewed literature contain overstatements that should be toned down. For example, “these results suggest that astrocytes can play a crucial role in 125 iron accumulation and redistribution occurring in FRDA”, lines 127-129. The majority of studies quoted in this section are not showing FRDA data.

6.    Citations are missing for some statements (e.g. line 127-129).

7.    I suggest introducing Table 1 earlier in the text, not at the end of the manuscript (in the conclusion´s section), as it introduces reviewed results. The Table´s organization is confusing in some parts. For example, it is not clear that the row/column referring to: Increased cerebellar COX2, is referring to the YG8R mice until you check it corresponds to Ref 40.

8.    Likewise, I suggest introducing Figure 1 earlier in the text.

9.    It could be interesting to include some (if any exists) human data/studies in which glial parameters, i.e. biomarkers, have been evaluated.

10.To strengthen the data reviewed in section 3.2., please consider including a recent paper showing the effects of SAG on mitochondrial dysfunction induced by the lack of frataxin on astrocytes (doi: 10.1186/s12974-022-02442-w).

Author Response

We thank this Reviewer for the opportunity to improve our manuscript. We revised the manuscript by responding point to point to all questions raised and evidencing all the changes and the additions in yellow throughout the text.

1.The manuscript must be reviewed for English language grammar and style, as some errors were present, and some parts are difficult to read/understand. 

We thank the Reviewer for the suggestion. We have now thoroughly revised the entire manuscript with a native speaker and marked the main modified sentences in yellow. See, for instance lines 60-64, 88-90, 102-106, 186-191, 193-199, 208-212, 226-232, 252-256, 290-296, 305-307.

2. Please define acronyms at their first use (e.g. CNS)

We did, thank you.

3. Gene/protein nomenclature should be used following standard rules. i.e. FXN gene, FXN protein.

We did, thank you.

4. Please use acronyms homogeneously in the manuscript, i.e. line 47, frataxin level, and line 48, FXN deficiency.

We checked and uniformed all the acronyms, thank you.

5. Some conclusions derived from the reviewed literature contain overstatements that should be toned down. For example, "these results suggest that astrocytes can play a crucial role in 125 iron accumulation and redistribution occurring in FRDA", lines 127-129. The majority of studies quoted in this section are not showing FRDA data.

As suggested, to avoid overstatements, we modified the Abstract and some sentences throughout the body text:

 "Therefore, these results suggest that astrocytes may participate in iron accumulation and aberrant redistribution in other disorders, including FRDA" (lines 120-121)

And "….increased mitochondrial iron content that favors oxidative stress and superoxide production, contributing to the non-cell-autonomous pathological process in FRDA" (lines 259-260)

"…suggesting a role of astrocytes in the progression of the disease…" (lines 276-277)

"Altogether, this evidence suggests that the expression of mutant frataxin in glial cells may act as a trigger, responsible for their reprogramming and functional impairment, contributing to the degeneration of the nearby neurons" (lines 227-329)

6. Citations are missing for some statements (e.g. line 127-129).

We added citations in different parts of the manuscript (lines 52, 80,123,124, 218, 246, 301)   

7. I suggest introducing Table 1 earlier in the text, not at the end of the manuscript (in the conclusion's section), as it introduces reviewed results. The Table's organization is confusing in some parts. For example, it is not clear that the row/column referring to: Increased cerebellar COX2, is referring to the YG8R mice until you check it corresponds to Ref 40.

As suggested, we have now provided a new Table 1 and moved it in paragraph 2 (line 125) and cited as follows: "Evidence of the involvement of neuroinflammatory mechanisms in FRDA has been detected in patients and disease models and is summarized in Table 1". 

 8. Likewise, I suggest introducing Figure 1 earlier in the text.

We introduced Figure 1 in paragraph 3 (line 154) and cited it as follows: "Specific pathways related to neuroinflammation are altered in microglia, astrocytes and myelinating glial cells in FRDA shown in Figure 1 and reported in detail in the following paragraphs".

9. It could be interesting to include some (if any exists) human data/studies in which glial parameters, i.e. biomarkers, have been evaluated.

We thank the Reviewer for this valuable suggestion. We added the new data on glia biomarkers, reported in the paper by Zeitlberger and colleagues, Front cell Neurosci, 2018:

"Consistently, the levels of plasma glial fibrillary acidic protein (GFAP) are significantly higher in FRDA patients, potentially reflecting glial activation" (lines 241-243)

10. To strengthen the data reviewed in section 3.2., please consider including a recent paper showing the effects of SAG on mitochondrial dysfunction induced by the lack of frataxin on astrocytes (doi: 10.1186/s12974-022-02442-w).

 We thank the Reviewer for this valuable suggestion. We added the following paragraph in section 3.2, lines 297-301: "It was recently demonstrated that targeting Sonic Hedgehog (SHH) with the Smoothened antagonist (SAG) rescued mitochondrial dysfunction and reverted the neurotoxicity induced by the lack of FXN in human astrocytes in vitro, showing the potential of pharmacologically targeting astrocytes cells to attenuate neurodegeneration in FRDA."

Reviewer 2 Report

see word-file

Author Response

We thank this Reviewer for the opportunity to improve our manuscript. We revised the manuscript by responding point to point to all questions raised and evidencing all the changes and the additions in yellow throughout the text.

The topic of this review "Neuroinflammation in Friedreich's Ataxia" is relevant in terms of putting another piece of the puzzle in understanding the pathophysiology in FRDA. Unfortunately, some passages are quite difficult to understand as a reader, partly due to too long and unnecessarily complex sentences and partly due to inappropriate wording. The authors might check once more the readability of their manuscript and shorten some sentences or making two out of one. 

We thank the Reviewer for the suggestion. We have now thoroughly revised the entire manuscript with a native speaker and marked main modified sentences in yellow. See, for instance lines 60-64, 88-90, 102-106, 186-191, 193-199, 208-212, 226-232, 252-256, 290-296, 305-307.

Furthermore I would add according two references – one on page 2, line 81-84, the other one page 5, line 197- 199.

We added the new references, thank you. 

In the conclusion section the first sentence that FRDA is among the most devastating neuromuscular disease is definitely not true from a clinical point of view. Of course it is a devastating disease – out of question – but stating that it is amongst the most devastating is wrong (just think of ALS, Duchenne muscular dystrophy, SMA,…) – besides, the mentioned "intense emotional effect on society" is not clear to me (keeping in mind that the majority of society does not even know the disease, being a rare disease above all).

As correctly suggested, we have now modified the first sentences of the paragraph as follows:" FRDA is a chronic, progressive and frequently life-threatening neuromuscular disease. Like all rare diseases, it has a high impact on the quality of life of patients and their families, accompanied by the lack of public awareness." (lines 334-336).